# Treatment of Far-Advanced Otosclerosis: Stapedotomy Plus Hearing Aids to Maximize the Recovery of Auditory Function—A Retrospective Case Series

**DOI:** 10.3390/healthcare11050676

**Published:** 2023-02-24

**Authors:** Giampietro Ricci, Salvatore Ferlito, Valeria Gambacorta, Mario Faralli, Pietro De Luca, Alfredo Di Giovanni, Arianna Di Stadio

**Affiliations:** 1Otolaryngology Department, University of Perugia, 06129 Perugia, Italy; 2Otolaryngology Department, San Giovannni-Addolorata Hospital, 00100 Rome, Italy; 3GF Ingrassia Department, Otolaryngology, University of Catania, 95124 Catania, Italy

**Keywords:** otosclerosis, hearing loss, advanced otosclerosis, auditory threshold, word recognition, speech, pure tone auditory test, stapedectomy, hearing aids, cochlear implant

## Abstract

Far-advanced otosclerosis (FAO) refers to severe otosclerosis with scarce auditory functions. The identification of the best method to correctly listen to sound and speech has a large impact on patients’ quality of life. We retrospectively analyzed the auditory function of 15 patients affected by FAO who were treated with stapedectomy plus hearing aids independent of the severity of their auditory deficit before surgery. The combination of surgery and hearing aids allowed excellent recovery of the perception of pure tone sounds and speech. Four patients, because of poor auditory thresholds, needed a cochlear implant after stapedectomy. Despite being based on a small sample of patients, our results suggest that stapedotomy plus hearing aids could improve the auditory capacities of patients with FAO independent of their auditory thresholds at T0. The careful selection of patients is fundamental to obtain the best outcomes.

## 1. Introduction

Otosclerosis defines the excessive bone resorption that affects the labyrinth capsule followed by the re-deposition of new sclerotic bone [1]. Commonly, the new bone growth around the oval window blocks the footplate of the stapes and causes conductive hearing loss. The otosclerotic process affects the cochlea and its structures (spiral ganglia and nerve fibers) in 10% of patients, causing mixed hearing loss; with the progression of the otosclerotic process (retrofenestral), hearing loss becomes profound and fully sensorineural [2].

Severe–profound sensorineural hearing loss (SNHL) caused by otosclerosis is named far-advanced otosclerosis (FAO).

In 1961, House and Sheeny [3] were the first to describe FAO as a disease characterized by an air conduction threshold of at least 85 dB without an identifiable bone conduction threshold. Today, it has been clarified that speech perception (word recognition score (WRS)) must be evaluated because the ability to hear sounds is not meaningful without a correct understanding of speech. In particular, patients with FAO, in which bone ossification could involve the modiolus (the part of the cochlea containing the spiral ganglia), could present speech discrimination thresholds discordant with those expected by the results of a pure tone auditory (PTA) test.

FAO can be treated using (i) conventional hearing aids, which due to the poor results that could be obtained, are only indicated in patients with contraindications to surgery [4]; (ii) stapedotomy plus conventional hearing aids; (iii) stapedotomy plus implantable middle-ear prostheses; and (iv) cochlear implants [5,6,7,8,9]. The latter allows patients to recover good hearing capacity, even in severe FAO, by adapting the surgery to the specific cases [10].

Any treatment has strengths, weaknesses, advantages, and disadvantages. Currently, despite the interesting proposal of Merkus et al. [5], the guidelines for the treatment of FAO are unclear, and the choice of the method to restore the hearing capacity is a physician’s personal decision. Some authors suggest stapedotomy followed by the application of hearing aids as an early treatment [1,8,11,12], while others propose cochlear implants as a first treatment instead of stapedotomy [11,13].

In 2011, Merkus et al. proposed a classification based on the score of the speech perception test (SPT) to address patients to the different methods of hearing rehabilitation [5]; he suggested that patients with SPT scores >50% should undergo stapedectomy, those with scores between 30 and 50 could be treated using stapedectomy or cochlear implants, and those with scores <30 need cochlear implants. This indication is not universally followed by neuro-otologists.

Our study aims to evaluate the improvement in hearing capacities in a series of patients who underwent stapedectomy plus hearing aids independent of the scores obtained in the SPT; this aim could validate (or invalidate) Merkus’s suggestions.

## 2. Materials and Methods

This retrospective clinical study was conducted in the Otolaryngology Department of Santa Maria della Misericordia University Hospital, a tertiary referral center. All procedures were approved by the internal revision board (IRB) of the hospital and conducted in accordance with the ethical principles of the Declaration of Helsinki; no approval number was released, given of the retrospective design of the study, following national laws.

We revised our clinical records to identify patients who were affected by FAO by screening the department data set from January 2014 to December 2020. In this period, 472 patients underwent stapedotomy surgery; 55 cases were bilaterally treated, for a total of 523 operations. In total, 15 interventions were performed on patients affected by FAO.

The following data were extracted from the clinical records: otoscopy findings, pure tone auditory test (PTA) and speech perception test (pre- and post-surgery) results, impedance, vestibular examinations, and the type of surgery performed. The surgical procedure performed on all 15 patients consisted of a primary stapedotomy on the worse-hearing ear.

All PTA tests were performed at our institution in the audiology unit before surgery (T0) and after 6 months (T1). From patients’ medical records, we collected data about the preoperative air conduction (AC) threshold (t0) (frequencies of 500, 1000, 2000, and 3000 Hz, according to the criteria of the American Academy of Otolaryngology Head and Neck Surgery Committee of Hearing and Equilibrium guidelines) [14], the speech perception test, the postoperative (6 months after surgery) air conduction threshold (T1) (frequencies of 500, 1000, 2000, and 3000 Hz), and the speech perception test. Then, pure tone averages (PTAv) were calculated, including frequencies of 500, 1000, 2000, and 3000 Hz.

The preoperative and postoperative speech perception tests were administered in an open field with bilateral hearing aids to obtain the open-set WRS at 60 dB HL. A list of simple bisyllabic words (see Figure 1) was used to test patients’ speech perception. This was a standard list of words used in Italy for adults called Bocca—Pellegrini. The results were considered normal with a WRS > 92%.

The SPT was measured by the average percentage, with values from 0% (indicating the total absence of perception) to >95% (meaning the full perception of all words).

Speech perception was considered satisfactory when the word recognition score recovery (T1 vs. T0) was more than 40%, good when it was between 39 and 21%, and poor when it was ≤20%. This sub-division was created considering the ability to understand speech in an open field in a normal speech interaction.

The selection of the ear that needed surgery was made based on the results of the audiological investigation. The ear with the worse auditory results underwent surgery.

### Statistical Analysis

Using one-way ANOVA and Bonferroni–Holmes (BH) ad hoc tests, we compared the single auditory frequency before and after stapedectomy. Two-tailed *t* tests (τ) were used to compare the PTAv pre- and post-treatment as well as the WRS at T0 and T1. A *p* value < 0.05 was considered statistically significant.

## 3. Results

We identified 15 patients (3.1% of patients undergoing stapedoplasty surgery), including 8 women and 7 men. The average age was 60 years (CI95%: 42–72). The diagnosis of FAO was confirmed during surgery. All these patients had postoperative audiological tests at 6 months; this observation time was considered a short follow-up.

All 15 patients had normal tympanic membranes before the operation, negative vestibular tests (no vertigo), and underwent the same surgical procedure (the insertion of a fluoroplastic piston prosthesis with a diameter of 0.4 mm and a variable length of 4–5 mm) performed by the same operator (GR).

All patients were prothesized with retroauricular high-power (BTE) hearing aids 21 days after stapedectomy; these prostheses used semi-linear power amplification technology (Phonak by SONOVA, Stäfa, Switzerland).

Statistically significant differences were observed comparing 500, 1000, 2000, and 3000 Hz before (T0) and after treatment (T1) (ANOVA: *p* < 0.0001). The improvement was statistically significant for 500 Hz (BH: *p* < 0.01), 1000 Hz (BH: *p* < 0.01), 2000 Hz (BH: *p* < 0.01), and 3000 Hz (BH: *p* < 0.01) (Figure 2).

The PTAv improved, with statistically significant values comparing T0 and T1 (τ: *p* < 0.0001). As shown in Table 1, the mean preoperative AC PTAv was 94.2 ± 5.2 dB HL (CI95%: 90–107.5), and the mean AC PTAv was 63.5 ± 17.9 dB HL (CI95%: 43.75–103.75) post-surgery (Figure 3).

Comparing the WRSs before and after treatment, we observed their improvement with a statistically significant value (τ: *p* < 0.0001). The average at T0 was 25% ± 13.5 (CI95%: 5–45%), while at T1 the mean was 62% ± 19.3 (CI95%: 25–80%) (Figure 4).

The average WRS recovery was 37 + 12.1 (CI95%: 15–55%); nine patients presented satisfactory results (recovery > 40%), four presented good results, and two presented poor results (Table 1).

Four cases presented postoperative problems: two had postoperative vertigo, which resolved within 48 h, and two cases suffered from dysgeusia, which resolved in a few weeks. Four patients (26.7%) were not satisfied with their hearing performance, despite a recovery over 30, because they had difficulties understanding people in normal, everyday conditions (noise and multi-voice) and had a WRS < 30%; for this reason, they underwent cochlear implant surgery 8–18 months after stapedectomy (bolded in Table 1). The patients were re-evaluated 6 months after surgery and presented WRSs over 65% in a free field (one 65%, one 75%, and two 90%).

## 4. Discussion

Overall, our results show that combining stapedectomy with hearing aids benefitted patients affected by FAO independent of their pre-surgery WRS. The degree of recovery was an important outcome, as was the recovery of specific auditory frequencies. In fact, the improvement in the hearing frequency over 1000 Hz positively impacted the WRS, as previously described [15]. By reducing the auditory thresholds at each frequency, stapedectomy and hearing aids also statistically improved the PTAv.

Despite being preliminary because we had a small sample of patients, our results have shown that Merkus’s algorithm for identifying the best treatment based on the WRS seems not to be feasible for all patients; in fact, despite treating all patients by combining stapedectomy and hearing aids independent of their WRS at T0, only four of the nine (44%) who had a score under 30% (candidates for cochlear implants following Merkus) needed a cochlear implant to recover good/satisfactory auditory functions.

Currently, several options are available for the treatment of FAO-related HL; thanks to the improvement of technology and surgical techniques, it is possible to perform cochlear implant surgery when necessary [11,16].

Stapedotomy presents some advantages compared to a cochlear implant. First, it is minimally invasive and can be performed under local anesthesia, a strength for treating elderly patients with comorbidities [17]. The prothesis used is cheaper than a cochlear implant device and allows better sound quality of the acoustic stimulation compared to electrical stimulation (especially when listening to music). Finally, the rehabilitation period is shorter than that required for a cochlear implant [18,19].

In our case series, 11/15 patients (73.3%) achieved satisfactory results in verbal communication, and in four cases the outcome was poor. In these unsuccessful cases, we used a cochlear implant, which allowed the patients to reach a good auditory ability. The literature shows that the success rate of stapedotomy in FAO ranges from 36 to 100% on tonal audiometry tests, with verbal recognition rates ranging from 38 to 75% [1,5]; these data are slightly lower than the results obtained with cochlear implants.

The presence of a very high or unvaluable bone conduction threshold does not always represent a negative prognostic factor, even if the success rate does not exceed 30% in those cases [11]. In patients with FAO, sensorineural hearing loss evidenced by PTA does not occur only due to the degeneration of hair cells. It is also partly caused the fixity of the annular ligament [7,19]. The latter was confirmed by Shea et al. (1999) [20], who reported that 42% of patients who did not have a preoperative bone conduction threshold showed a measurable threshold after stapedotomy surgery. Furthermore, patients with otosclerosis might show a subsequent deterioration of sensorineural hearing loss that cannot be explained by age alone [7,19,21]. However, it is important to highlight that performing a stapedotomy does not compromise or impact the results of a possible subsequent cochlear implant [22].

Common complications of stapedotomy are the further loss of residual hearing, tinnitus, dizziness, and dysgeusia (due to lesions of the chorda tympany). These adverse events were recently reported by Teaima et al. (2022) [18] in 18.6% of patients who underwent stapedotomy. This percentage is slightly higher than that of complications observed with cochlear implants (13.6%). In particular, dysgeusia, tinnitus, and residual hearing loss are more frequent during stapedotomy, while vertigo is more frequent with cochlear implants. In our study, two patients (13.3%) presented postoperative dizziness and two presented dysgeusia (13.3%); however, these symptoms spontaneously regressed in a few days and weeks, respectively. There are some factors that could predict the success of stapedotomy, such as the age of the patient, gender [19], the duration of hearing deprivation, and preoperative radiological evaluation [10]. None of these factors appeared to impact the results in a systematic way.

The use of a cochlear implant in the treatment of FAO allows patients to obtain better results compared to the conventional stapedotomy plus hearing aid with regards to the improvement in the hearing threshold. While some authors [23] found a positive correlation between postoperative verbal performance and the extension of the otosclerotic ossification, other studies showed postoperative verbal performance scores comparable to those of non-otosclerosis control groups [24,25]. This can be explained by the normal or slightly reduced number of spiral ganglia, even in the case of extensive cochlear ossification [26]. The literature reports verbal discrimination rates between 45 and 98% after cochlear implant surgery in patients with FAO [10].

Compared to stapedotomy, cochlear implants present some disadvantages, such as the higher cost of the device, greater surgical difficulties, higher incidence of complications (due to facial nerve risks during the surgical procedure), and greater post-implantation rehabilitation difficulties.

A cochlear implant surgical procedure could be challenging in FAO due to cochlea bone alterations caused by the otosclerotic processes. The degree of ossification ranges from 5 to 51%, depending on the study [23]. Bone ossification is more common in the basal turn of the cochlea, with possible involvement in the region of the round window; conversely, isolated ossifications of the middle or apical turns are very rare. In the case of cochlear ossification, some authors recommend using a subtotal petrosectomy rather than a posterior tympanotomy to obtain a better exposure of the obliterated round window [27,28]. The presence of ossifications or, on the contrary, the presence of areas of bone resorption can lead to an incomplete insertion or the mispositioning of the array [29]. Rotteveel et al. (2004) [23] reported electrode insertion issues in 10 cases out of 53 patients (18.8%). However, it has been shown that cochlear ossification, once the array is correctly placed, does not affect postoperative performance, even in long-term follow-up [28,29,30]. The decision to use flexible or stiffer arrays with a stylet to facilitate insertion can be chosen case by case [18].

One of the major complications of cochlear implant surgery in FAO is the stimulation of the facial nerve, which was reported by Rotteveel et al. to occur in 38% of cases [23]. Structural alterations, such as hyalinization of the spiral ganglion, could permit the transmission of the electrical stimulus to the Fallopian canal. In general, otosclerotic processes are believed to be the cause of low electrical resistance. The likelihood of this problem might also be influenced by the type of electrode used; for example, the perimodiolar type is considered better than the straight one to avoid this problem [31,32]. Furthermore, patients with important bone alterations identified with preoperative CT show higher incidence of indirect facial nerve stimulation. Facial nerve stimulation can usually be managed by reprogramming or deactivating the electrodes. However, when many electrodes must be deactivated, the performance of the device could be significantly reduced [32]. Hence, a careful preoperative evaluation of a patient’s findings (auditory test, speech perception, age, superior auditory function [33], and CT/MRI) should be conducted before selecting the method to use [34].

It must be considered that, even after the success of a cochlear implant surgery, the rehabilitation of an otosclerotic patient can be challenging. In fact the evolution of the osteo-thickening processes can affect the performance of the implant, requesting frequent reprogramming with higher stimuli [35]. For this reason, the selection of the patient’s ear is fundamental, especially in case of bilateral disease [36].

Teaima (2022) [16] showed that patients had better thresholds on PTA tests when treated with cochlear implants (29.1 dB) compared to stapedotomy (52.3 dB); however, the latter allowed better scores in verbal discrimination for monosyllables and bisyllabic words (34% and 56.6%) compared to cochlear implants (28.1 and 55.2%). Kabbara et al. (2018) [6] reported satisfactory outcomes in 60% of patients treated with stapedotomy and in 85% of patients treated with cochlear implants (word reception scores greater than 50%). Berrettini et al. [36] showed statistically better verbal discrimination abilities in patients treated with cochlear implants compared to stapedectomy.

Heining showed that stapes surgery can be efficient for restoring the auditory function in patients with FAO [37].

Today, there is a lack of guidelines to support hearing specialists in the choice between stapedotomy and cochlear implants for the treatment of patients with FAO. Many authors recommend starting with stapedotomy, followed by a cochlear implant in the case of failure; a stapedectomy is relatively simple to place and has a low cost [8].

In 2011, Merkus [5] proposed an interesting algorithm based on the evaluation of verbal recognition skills; a cochlear implant is recommended for recognition rates lower than 30%; for percentages between 30 and 50%, both stapedotomy and implant are recommended; and for percentages higher than 50%, stapedotomy surgery is recommended. If the air–bone gap (ABG) is considered, a cochlear implant is recommended for values below 30 dB HL.

By analyzing our case series following the Merkus algorithm, the 4 patients who subsequently received cochlear implants had percentages of verbal discrimination lower than 30%, but out of the 11 successfully operated cases, 3 had verbal recognition lower than 30% with a prosthesis.

In our study, we obtained stable and short-lasting results in 11 out of 15 patients. Four of them subsequently underwent a cochlear implant. The short follow-up is a significant weakness of our study as well as other previously conducted studies [38]; in fact, because surgery can only delay the progression of otosclerosis and does not cure it, a longer follow-up could show a worsening of the auditory capacities both in patients who underwent stapedectomy and those treated with cochlear implants due to the progression of the disease.

Long follow-ups at 12 and 24 months after the treatments could better underline the benefit of one technique or the other.

Lachance et al. (2012) [11] showed the results of stapedotomy in a group of 16 candidates for cochlear implantation: 87% of these patients obtained satisfactory results and were no longer considered for cochlear implantation. Heining et al. [37] reported that 10% of patients who underwent stapedotomy needed an additional treatment with a cochlear implant to obtain good auditory results. Some parameters must be considered during the selection phase and should guide the choice of the most suitable treatment for a patient; the results of tone and verbal audiometry, clinical history, comorbidities, radiological results, the duration of hearing deprivation, residual contralateral ear hearing, and patient preferences are useful to correctly address the choice of treatment.

*Limitations of the study:* The main limitation of this study is the retrospective design; prospective studies comparing the auditory results of patients with FAO treated with stapedectomy plus hearing aids versus cochlear implants should be performed to clarify the benefit of one technique or the other. Secondly, these patients had a very short follow-up, so the results could change after 12/24 months, and for this reason our findings should be considered preliminary results. Thirdly, we had a very small group of patients. Additionally, because surgery is routinely performed on the ear with worse auditory thresholds, the difference in the “good ear” among patients could have affected our results. Finally, the diagnosis of FAO was made during the operation, while, as shown by Messineo et al. [11], using radiological findings, it is possible to maximize the results of cochlear implant surgery with correct planning and the use of specific software that helps doctors perform the surgery [15].

## 5. Conclusions

Our case series showed that patients with FAO might benefit from stapedectomy plus hearing aids; this could allow them to recover good auditory functions in terms of both the PTA test and the SPT. A quarter of the patients needed additional surgery because the auditory results were unsatisfactory; due to the large impact of auditory abilities on cognition [39,40], the personal perception of the patients about their satisfactory or unsatisfactory hearing capacities must be considered. The use of cochlear implants allowed these patients to recovery their hearing capacity enough to be comfortable in the everyday lives. Both stapedotomy and cochlear implantation are effective therapies for FAO. The lower invasiveness and lower cost of stapedotomy makes it advisable as a first approach, but the treatment must be tailored in any case to the patient’s needs and expectations. Considering parameters such as audiological data, the clinical history and general condition of the patient, radiological data, the duration of hearing deprivation, and patient preferences can be helpful to plan a surgery correctly with the best understanding of a case to obtain satisfactory hearing rehabilitation.

## Figures and Tables

**Figure 1 healthcare-11-00676-f001:**
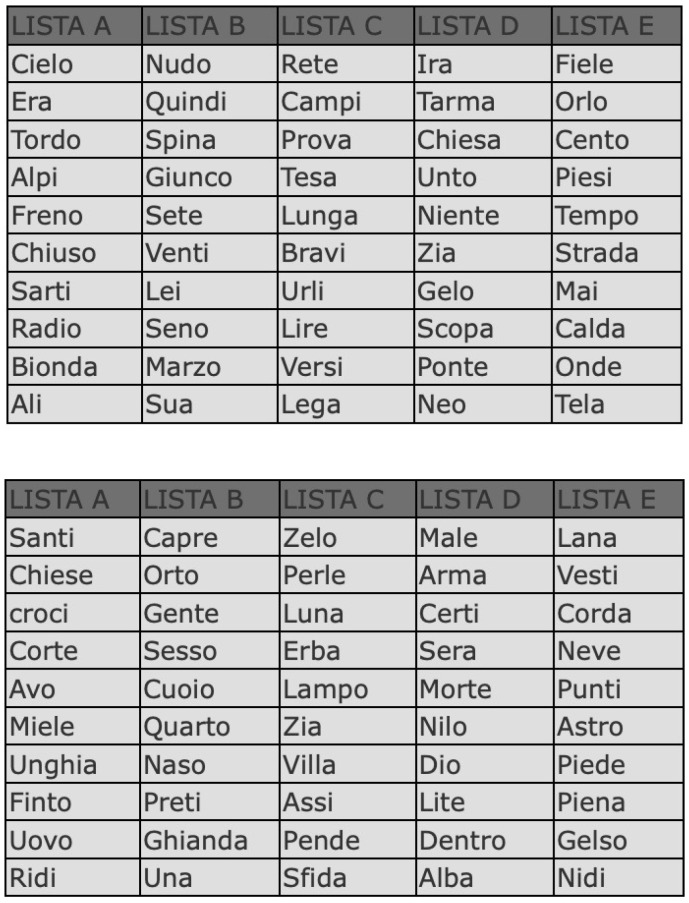
List of Italian words used to perform the speech perception test.

**Figure 2 healthcare-11-00676-f002:**
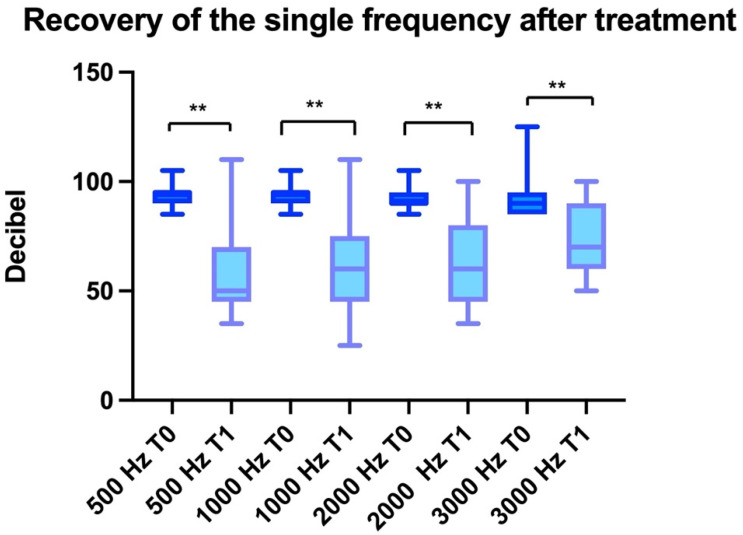
The improvement in the auditory thresholds of each frequency obtained after surgery, comparing the baseline with hearing aids and post-treatment (stapedectomy plus hearing aids). “**” indicates *p* < 0.01.

**Figure 3 healthcare-11-00676-f003:**
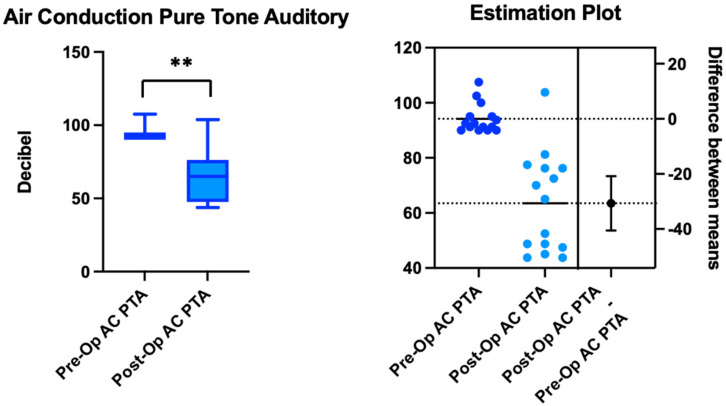
The results of the air conduction (AC) and pure tone average (PTA) tests before (Pre-Op) and after surgery (Post-Op). The estimation plot shows the differences between the means, and the “**” indicates statistical significance of *p* < 0.0001.

**Figure 4 healthcare-11-00676-f004:**
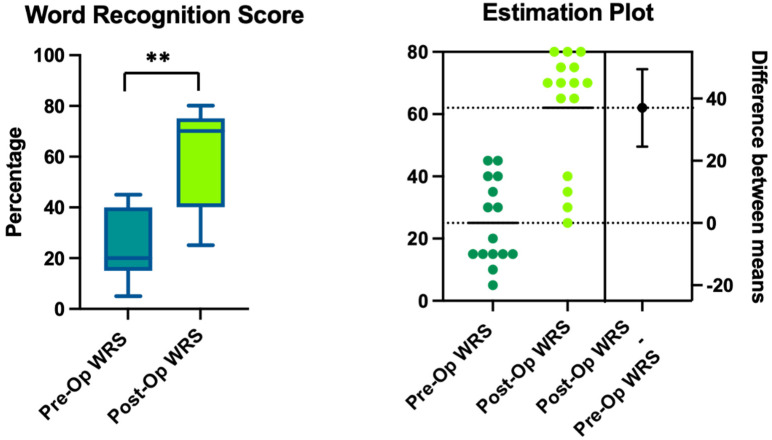
The results of the word recognition tests (word recognition score (WRS)) before (Pre-Op) and after surgery (Post-Op) at the 6-month follow-up. The higher values indicate the percentage of words recognized during the test; normal WRS scores are > 90%. The estimation plot shows the differences between the means, and the “**” indicates statistical significance of *p* < 0.0001.

**Table 1 healthcare-11-00676-t001:** Characteristics of our sample and details of auditory and speech understanding values before and after surgery. Bolded rows represent patients who needed cochlear implants due to unsatisfactory recovery of auditory functions when combining stapedectomy and hearing aids.

Patient	Gender	Age	Side	AC-PTA (dB HL) T0	AC-PTA (dB HL) T1	WRS T0	WRS T1	WRS Gain
1	M	65	Right	91.25	43.75	40	80	40
2	M	42	Left	100	72.5	30	70	40
3	F	64	Left	92.5	65	15	70	55
4	F	61	Left	90	43.75	40	80	40
5	M	51	Left	91.25	48.75	35	75	40
6	M	65	Left	107.5	76.25	15	65	50
**7**	**F**	**52**	**Left**	**90**	**76.25**	**15**	**40**	**25**
8	F	58	Right	92.5	52.5	45	75	30
9	M	57	Right	95	47.5	15	70	55
10	F	64	Right	90	70	20	65	45
**11**	**M**	**48**	**Left**	**90**	**81.25**	**10**	**25**	**15**
**12**	**F**	**64**	**Right**	**91.25**	**77.5**	**15**	**35**	**20**
13	F	70	Right	95	45	45	80	35
**14**	**M**	**68**	**Right**	**102.5**	**103.75**	**5**	**30**	**25**
15	F	72	Right	93.75	48.75	30	70	40

## Data Availability

Data are available upon written request from the first author.

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
