# Peer review of "Treatment of Far-Advanced Otosclerosis: Stapedotomy Plus Hearing Aids to Maximize the Recovery of Auditory Function—A Retrospective Case Series"

_healthcare, 2023, doi:10.3390/healthcare11050676_

Round 1

Reviewer 1 Report

The paper claims to compare two primary interventions for Far Advanced Otosclerosis (FAO): (1) stapedotomy plus hearing aids and (2) cochlear implant (CI). The retrospective analysis included 15 subjects with FAO that received stapedotomy plus hearing aids. The paper reveals significant improvement of the AC PTA and WRS 6 months postoperative compared to preoperative. Four of the 15 subjects received a CI after the stapedotomy because of “unsatisfactory results”. The authors conclude that stapedotomy and CI are effective treatments for FAO and advice a personalized approach.

The paper states that they compared two primary interventions for FAO. In reality, the authors compared hearing outcomes before and after (1) stapedotomy plus hearing aids. The authors then state that four patients received a CI in a second intervention. They can therefore not concluded, that both treatments are effective.

The paper lacks methodically and in discussion and conclusion of results. The writing is hard to follow in some places. An English revision by a native speaker is advised.

Introduction:

-          Please rewrite and expend on “Currently it is necessary to also consider verbal discrimination as additional concern of this condition”

-          Please correct and rewrite “Each treatment presents advantages, complication […]

-          The purpose stated by the authors suggest, that they compared hearing outcomes between primary treatment with (1) stapedotomy plus hearing aids or (2) CI. This is not the case.

Materials and Methods:

-          Please state the speech perception test used and reference it.

-          Please specify patient criteria concerning the contralateral side.

-          Why did the authors use the Wilcoxon Signed-Rank Test? Did the authors check for lack of normal distribution?

Results:

-          Data of AC PTA4 and WRS are stated appropriately and displayed nicely in the boxplots.

-          Figure 3 does not add any new information.

-          How is an “unsatisfactory results” defined?

-          How did the authors test for “communicate satisfactorily face-to-face”?

-          Did the hearing capacity of the contralateral side influence the results?

-          Which are the four patients that received a CI? Are there factors differentiating them from the 11 that were “satisfied” with the hearing?

-          How did the four CI-users performed with their CI? Was the results “satisfactory”?

Discussion:

The discussion is poorly written and lacks sufficient references. The authors do not discuss the results of the presents study or put them in context to other studies.

-          The authors greatly focus on the cost of surgeries. However, no actual costs or cost benefit analysis are discussed. The necessity of two interventions (stapedotomy and CI) are not even mentioned.

-          The algorithm by Merkus et al. (2011) and its application on the present study are mentioned for the first time in the discussion. Please give this information in methods and materials as well as results.

-          The interesting results from the application of the algorithm by Merkus et al. (2011) are not discussed sufficiently. No conclusions are drawn from this.

-          The authors claim “long-lasting effects” but only report a follow-up of 6 months.

Conclusion:

The authors conclude that CI and stapedotomy are effective treatment for FAO. This conclusion cannot be based on the present data, as Ricci et al. did not compare CI and stapedotomy and reported no outcome with CI. Even the effectiveness of a stapedotomy is hard to draw from only 15 subjects.

Author Response

Dear Editors and Reviewers,

Thank you for the thoughtful review of our manuscript and for sharing your valuable comments. We are grateful for the opportunity re-submit a revised manuscript for your consideration. The critiques have allowed us to strengthen the manuscript and to address several areas requiring clarification. We have carefully read all the comments and revised the manuscript accordingly.

Below, we have responded to each comment point-by-point, presenting the reviewer critiques in regular type and our responses in bolded italics. Excerpts of revised text are also provided (indented text). In the main document, we have used track changes and red to show edits.

Thank you again for your time and insightful review of our paper.

Reviewer 1

The paper claims to compare two primary interventions for Far Advanced Otosclerosis (FAO): (1) stapedotomy plus hearing aids and (2) cochlear implant (CI). The retrospective analysis included 15 subjects with FAO that received stapedotomy plus hearing aids. The paper reveals significant improvement of the AC PTA and WRS 6 months postoperative compared to preoperative. Four of the 15 subjects received a CI after the stapedotomy because of “unsatisfactory results”. The authors conclude that stapedotomy and CI are effective treatments for FAO and advice a personalized approach. 

The paper states that they compared two primary interventions for FAO. In reality, the authors compared hearing outcomes before and after (1) stapedotomy plus hearing aids. The authors then state that four patients received a CI in a second intervention. They can therefore not concluded, that both treatments are effective. 

Thank you for this important comment that has allowed us to re-write correctly our paper. You are right the poor English created a miss-understanding. In the current version we clarified that we did not compare the outcome of the patients with cochlear implants and stapedectomy plus hearing device. All patients performed stapedectomy and hearing device as treatment of FAO, however 4 of them needed to undergo additional surgery (8 months after the stapedectomy) to insert cochlear implant due to the unsatisfactory recovery of the auditory functions.

The paper lacks methodically and in discussion and conclusion of results. The writing is hard to follow in some places. An English revision by a native speaker is advised.

The paper has been totally re-written and the English revised by a native speaker. The material and methods have been revised, enriched with additional info that can allow the reproducibility of our work. Moreover new statistically analyses have be done and the results and the discussion been changed following the new obtained results.

Introduction:

-          Please rewrite and expend on “Currently it is necessary to also consider verbal discrimination as additional concern of this condition”

We changed it as suggested.

-          Please correct and rewrite “Each treatment presents advantages, complication […]

We changed it as suggested.

-          The purpose stated by the authors suggest, that they compared hearing outcomes between primary treatment with (1) stapedotomy plus hearing aids or (2) CI. This is not the case. 

You are right. We totally re-wrote the paper, including the aim and the material and method, as well as the title so to correctly refer the content of the article.

Materials and Methods:

-          Please state the speech perception test used and reference it.

We added this info and also a table containing the words used for the test. The table is in Italian because the study has been conducted on Italian mother language patients.

-          Please specify patient criteria concerning the contralateral side. 

This info has been added in this revision.

-          Why did the authors use the Wilcoxon Signed-Rank Test? Did the authors check for lack of normal distribution?

The statistical analyses have been revised by the last author that have a master in Bio-statistic. New statistic using one-way ANOVA-more appropriate test-has been done and the results re-written.

Results:

-          Data of AC PTA4 and WRS are stated appropriately and displayed nicely in the boxplots.

Thank you.

-          Figure 3 does not add any new information.

We removed it.

-          How is an “unsatisfactory results” defined? 

We defined how satisfactory/unsatisfactory results have been identified in the material and method section.

-          How did the authors test for “communicate satisfactorily face-to-face”?

We remove this sentence; in this version the correct method of evaluation has been explained.

-          Did the hearing capacity of the contralateral side influence the results?

We did not consider this, but we added this point in the study limitation section that we added in this revised paper.

-          Which are the four patients that received a CI? Are there factors differentiating them from the 11 that were “satisfied” with the hearing?

We clarified this point. Now the patients who needed CI have been bolded in table 1. These patients were the ones whose auditory recovery was poor and unsatisfactory.

-          How did the four CI-users performed with their CI? Was the results “satisfactory”?

This revised version clarified that none of the patients with cochlear implant were evaluated in this paper, we only reported the results of patients who combined surgery and hearing aids.

Discussion:

The discussion is poorly written and lacks sufficient references. The authors do not discuss the results of the presents study or put them in context to other studies. 

We re-wrote the discussion even considering the new results obtained. Now our results are discussed as first and then the literature. Moreover, a series of references have been added.

-          The authors greatly focus on the cost of surgeries. However, no actual costs or cost benefit analysis are discussed. The necessity of two interventions (stapedotomy and CI) are not even mentioned.

This topic has been discussed in this revision.

-          The algorithm by Merkus et al. (2011) and its application on the present study are mentioned for the first time in the discussion. Please give this information in methods and materials as well as results.

Thank you for this important comment. We added this concept in the intro and we revised this article keeping Merkus work in the center of our aim, as well as discussing our results based on the differences that we observed compared to the Merkus results.

-          The interesting results from the application of the algorithm by Merkus et al. (2011) are not discussed sufficiently. No conclusions are drawn from this.

We did it in this revision.

-          The authors claim “long-lasting effects” but only report a follow-up of 6 months. 

You are right we changed this word, and we added the short follow-up as limitation of the study.

Conclusion:

The authors conclude that CI and stapedotomy are effective treatment for FAO. This conclusion cannot be based on the present data, as Ricci et al. did not compare CI and stapedotomy and reported no outcome with CI. Even the effectiveness of a stapedotomy is hard to draw from only 15 subjects.

All the article has been revised, conclusion and title included. Our results are only preliminary and could only suggest, not confirm.

Reviewer 2 Report

Far advanced otosclerosis – FAO – is a fairly uncommon condition. Otosclerotic foci may extend deeper into the labyrinth, resulting in retrofenestral otosclerosis and severe mixed hearing loss. The degree of hearing loss is characterized as an air conductive threshold of less than 85 dB with a non-evaluable bone conduction threshold. In most patients stapedotomy combined with hearing aids will generate good hearing results but more recently treatment by cochlear implants have also yielded good hearing results. Stapedotomy compared with cochlear implant presents some advantages as lower cost of the device, less surgical difficulties and complications and an easier and faster post-surgery rehabilitation. Still there are no universal guidelines available for otosurgeons concerning which treatment modality would be most useful for the individual patient.

The present manuscript describes a study on stapedotomy on FAO patients. 15 patients, from 2014 to 2020, were evaluated by otoscopy, PTA, speech perception test, impedance and vestibular examination. The diagnosis of FAO was confirmed during the surgery. Post-operative evaluation was performed 6 months after surgery. PTA and WRS improved in 11 patients, whereas the improvement was minimal in 4 patients. These 4 patients were later subjected to cochlear implants. The authors conclude that both stapedotomy and cochlear implantation are affective therapies for FAO, but suggest stapedotomy as a first approach being less invasive and of a lower cost.

This is another retrospective study on a limited number of patients. Despite being subjected to adequate audiological tests, with results indicating FAO, the patients were not evaluated by additional radiology and the diagnosis thus confirmed during surgery. The follow up were performed 6 months after the surgery, which is a quite short time interval. One would have liked to have some long-term results which might more adequately show the benefit of one treatment procedure in comparison to another. 

Though the study is well described and with an interesting Discussion and adequate references I cannot see that this study contributes further to our understanding of the management of FAO.  Against this background I cannot recommend this manuscript for publication in Healthcare. A well-designed prospective study on stapedotomy and cochlear implants in FAO patients is needed, though it should be time-consuming and with several clinics participating in such a study. It might well be that there is not either or between the two treatment strategies and that instead the choice of method has to be individualized.

Author Response

Dear Editors and Reviewers,

Thank you for the thoughtful review of our manuscript and for sharing your valuable comments. We are grateful for the opportunity re-submit a revised manuscript for your consideration. The critiques have allowed us to strengthen the manuscript and to address several areas requiring clarification. We have carefully read all the comments and revised the manuscript accordingly.

Below, we have responded to each comment point-by-point, presenting the reviewer critiques in regular type and our responses in bolded italics. Excerpts of revised text are also provided (indented text). In the main document, we have used track changes and red to show edits.

Thank you again for your time and insightful review of our paper.

Reviewer 2

Far advanced otosclerosis – FAO – is a fairly uncommon condition. Otosclerotic foci may extend deeper into the labyrinth, resulting in retrofenestral otosclerosis and severe mixed hearing loss. The degree of hearing loss is characterized as an air conductive threshold of less than 85 dB with a non-evaluable bone conduction threshold. In most patients stapedotomy combined with hearing aids will generate good hearing results but more recently treatment by cochlear implants have also yielded good hearing results. Stapedotomy compared with cochlear implant presents some advantages as lower cost of the device, less surgical difficulties and complications and an easier and faster post-surgery rehabilitation. Still there are no universal guidelines available for otosurgeons concerning which treatment modality would be most useful for the individual patient.

The present manuscript describes a study on stapedotomy on FAO patients. 15 patients, from 2014 to 2020, were evaluated by otoscopy, PTA, speech perception test, impedance and vestibular examination. The diagnosis of FAO was confirmed during the surgery. Post-operative evaluation was performed 6 months after surgery. PTA and WRS improved in 11 patients, whereas the improvement was minimal in 4 patients. These 4 patients were later subjected to cochlear implants. The authors conclude that both stapedotomy and cochlear implantation are affective therapies for FAO, but suggest stapedotomy as a first approach being less invasive and of a lower cost.

This is another retrospective study on a limited number of patients. Despite being subjected to adequate audiological tests, with results indicating FAO, the patients were not evaluated by additional radiology and the diagnosis thus confirmed during surgery. The follow up were performed 6 months after the surgery, which is a quite short time interval. One would have liked to have some long-term results which might more adequately show the benefit of one treatment procedure in comparison to another. 

Though the study is well described and with an interesting Discussion and adequate references I cannot see that this study contributes further to our understanding of the management of FAO.  Against this background I cannot recommend this manuscript for publication in Healthcare. A well-designed prospective study on stapedotomy and cochlear implants in FAO patients is needed, though it should be time-consuming and with several clinics participating in such a study. It might well be that there is not either or between the two treatment strategies and that instead the choice of method has to be individualized.

Thank you for your exhaustive comments. We totally re-wrote this paper. The aim was better focused, the data were re-analyzed, and new results were identified. We specified that we did not compare cochlear implant and stapedectomy plus hearing aids, but all our patients were treated by stapedectomy plus hearing aids as first line treatment, and then in presence of unsatisfactory results they underwent cochlear implant surgery after 8 months. The current version explains that despite we performed in all case stapedectomy plus hearing aids without following Merkus algorithm we have obtained satisfactory auditory results even in those patients in whom the expectation should be poor results and that should undergo cochlear implant following Merkus algorithm. We added a section with the study limitations-including the small number of patients, the short follow-up, the lack of radiologic diagnosis and other important limitations.

We concluded by saying that our study can only suggest and not affirm that stapedectomy plus hearing aids must be performed for FAO. We also underlined the need of prospective comparative study to identify the gold standard.

We hope that this new version is more convincing and you might reconsider the value of our work.

Reviewer 3 Report

Hello

Dear authors, you have touched a very important aspect about the otosclerosisTreatment of Far Advanced Otosclerosis: stapedotomy or cochlear implant to maximize the recovery of the auditory function. A retrospective case-series study. 
The study is well structured and the results are well analyzed. I suggest the correction of a write error at the 20th line of the introduction (advantages and complications) and also in the fifth line of results (decrease is correct).

Congrats

Author Response

Dear Editors and Reviewers,

Thank you for the thoughtful review of our manuscript and for sharing your valuable comments. We are grateful for the opportunity re-submit a revised manuscript for your consideration. The critiques have allowed us to strengthen the manuscript and to address several areas requiring clarification. We have carefully read all the comments and revised the manuscript accordingly.

Below, we have responded to each comment point-by-point, presenting the reviewer critiques in regular type and our responses in bolded italics. Excerpts of revised text are also provided (indented text). In the main document, we have used track changes and red to show edits.

Thank you again for your time and insightful review of our paper

Reviewer 3

Dear authors, you have touched a very important aspect about the otosclerosis: Treatment of Far Advanced Otosclerosis: stapedotomy or cochlear implant to maximize the recovery of the auditory function. A retrospective case-series study. 

The study is well structured and the results are well analyzed. I suggest the correction of a write error at the 20th line of the introduction (advantages and complications) and also in the fifth line of results (decrease is correct).

Thank you we corrected the mistake.

Round 2

Reviewer 1 Report

The authors re-wrote large parts of the manuscript and have thereby improved the manuscript greatly.

Introduction

(1)   "Today, it has been clarified that speech perception (Word Recognition Score,WRS) is an important parameter to evaluate."

This "expansion" on the original sentence "Currently it is necessary to also consider verbal discrimination as additional concern of this condition” is insufficient. Please state why WRS is necessary to be reported and back your claim up with literature. To clarify possible misunderstandings: in the opinion of the reviewer WRS is necessary. However the authors make a weak or no case why this is.

(2)   “FAO can be treated i) by conventional hearing aids, which generally offer poor results and are indicated only in patients with contraindications to surgery; […]”.

Please provide references for the claim, that conventional hearing aids are only indicated in patients with contraindications to surgery and the poor results.

(3)   “Currently, despite the proposal of Merkus et al [ref], the guidelines for treating FAO are not clear and the method for restoring of hearing capacity is a physician’s personal decision.”

Please include all intended references [ref] before submitting or re-submitting a manuscript.

(4)   “Merkus et al. in 2011 proposed a classification based on the score of speech perception test (SPT) to address the patients to different methods of hearing rehabilitation; he suggested that > 50% patients should undergo stapedectomy, between 30 and 50 to stapedectomy or cochlear implant and < 30 to cochlear implant. Anyway, this indication is not universally followed by the neuro-otologist.”

Please clarify that the percentages confer to the SPT results.

Methods

(1)   Please state the name of the applied speech test. Has this speech test been published in Italian speaking literature? Is it a test used through our Italian speakers? Are the words provided in the table monosyllabic? What are the expected test results in subjects with normal hearing? 100%?

(2)   “Speech perception was considered satisfactory when word recognition score gain (T1 versus T0) was over > 40%, good when it was between 39 and 21% and poor < 20%.”

Is the definition for “satisfactory” based on any literature or performances (e.g. ability to use a telephone)?

(3)   The statistical analysis was redone and improved.

Results

(1)   “All patients had normal tympanic membranes, negative vestibular test (no vertigo) and […]”

Is this data preoperative? Please include this information.

(2)   How long after operation were the subjects fitted with a hearing aid? Is the T1 measurement with 6 months experience with hearing aids?

(3)   “Statistically significant differences were observed comparing 500, 1000, 2000 and 4000 Hz before (T0) and after treatment (T1) (ANOVA: p <0.0001).”

Based on the rest of the manuscript 3 kHz was used. Please correct.

(4)   Table 1. Please clarify in the Table header, why subjects are bolded.

(5)   “Four patients (26.7%), were not satisfied about their hearing performance despite a recovery over 30, […]”

Were the subjects not satisfied subjectively? Based on audiological outcome? Please elaborate.

(6)   Please provide outcome with CI.

Discussion

(1)   “By reducing the thresholds of each frequency, stapedectomy statistically improved the PTAv also.”

Please add “stapedectomy and hearing aids”.

(2)   Please be careful in proposing conclusions from you study.

(3)   “One of the major complications of cochlear implant surgery in FAO is the stimulation of the facial nerve which is reported up to 38% [16,30,31].”

Please give the range reported in the literature.

(4)   “The short follow-up is a strong weakness in our study, as well as in other studies previously done [ref];”

Please include all intended references [ref] before submitting or re-submitting a manuscript.

(5)   Adding the limitations of their study and possible future studies needed improves the manuscript greatly.

Author Response

Dear Reviewer,

Thank you for the thoughtful review of our manuscript and for sharing your valuable comments. We are grateful for the opportunity to submit a revised manuscript for your consideration. The critiques have allowed us to strengthen the manuscript and to address several areas requiring clarification. We have carefully read all the comments and revised the manuscript accordingly.

Below, we have responded to each comment point-by-point, presenting the reviewer critiques in regular type and our responses in bolded italics. Excerpts of revised text are also provided (indented text). In the main document, we have used track changes and red to show edits.

Thank you again for your time and insightful review of our paper

The authors re-wrote large parts of the manuscript and have thereby improved the manuscript greatly.

Thank you for your positive feedback.

Introduction

(1)   "Today, it has been clarified that speech perception (Word Recognition Score,WRS) is an important parameter to evaluate." 

This "expansion" on the original sentence "Currently it is necessary to also consider verbal discrimination as additional concern of this condition” is insufficient. Please state why WRS is necessary to be reported and back your claim up with literature. To clarify possible misunderstandings: in the opinion of the reviewer WRS is necessary. However the authors make a weak or no case why this is.

We underlined that is fundamental and we explain why. The new sentence is “Today, it has been clarified that speech perception (Word Recognition Score, WRS) is a fundamental parameter to evaluate because the only ability of listening to the sounds is not meaningful that patients can correctly understand speech. Patients with FAO in particular, in which bone ossification could involve the modiolus - part of the cochlea containing the spiral ganglions-, could present speech discrimination test thresholds not overlapping the expected one following the results of pure tone auditory (PTA) test.  “

(2) “FAO can be treated i) by conventional hearing aids, which generally offer poor results and are indicated only in patients with contraindications to surgery; […]”. 

Please provide references for the claim, that conventional hearing aids are only indicated in patients with contraindications to surgery and the poor results.

We added the missed reference.

(3)   “Currently, despite the proposal of Merkus et al [ref], the guidelines for treating FAO are not clear and the method for restoring of hearing capacity is a physician’s personal decision.” 

Please include all intended references [ref] before submitting or re-submitting a manuscript.

We added the missed references.

(4)   “Merkus et al. in 2011 proposed a classification based on the score of speech perception test (SPT) to address the patients to different methods of hearing rehabilitation; he suggested that > 50% patients should undergo stapedectomy, between 30 and 50 to stapedectomy or cochlear implant and < 30 to cochlear implant. Anyway, this indication is not universally followed by the neuro-otologist.”

Please clarify that the percentages confer to the SPT results. 

Thank you for this note. We clarified that the percentage refers to SPT

Methods

  • Please state the name of the applied speech test. Has this speech test been published in Italian speaking literature? Is it a test used through our Italian speakers? Are the words provided in the table monosyllabic? What are the expected test results in subjects with normal hearing? 100%?

We added this relevant info. “A list of simple bisyllabic words (see figure 1) was used in place of the pulse tone. This is a standard list of words used in Italy to test adults called Bocca – Pellegrini. The results is considered normal with a WRS > 92%.”

(2)   “Speech perception was considered satisfactory when word recognition score gain (T1 versus T0) was over > 40%, good when it was between 39 and 21% and poor < 20%.” 

Is the definition for “satisfactory” based on any literature or performances (e.g. ability to use a telephone)?

We clarified that the subdivision has been done as follow ” This sub-division was done considering the capacity of understanding the speech in open field in a normal speech interaction”

(3)   The statistical analysis was redone and improved.

Thank you

Results

(1)   “All patients had normal tympanic membranes, negative vestibular test (no vertigo) and […]” 

Is this data preoperative? Please include this information.

We clarified that the data were pre-operative.

(2)   How long after operation were the subjects fitted with a hearing aid? Is the T1 measurement with 6 months experience with hearing aids?

Patients started using HA three week after surgery. After this we waited 6 months to re-evaluate the patients’ auditory capacities.

(3)  “Statistically significant differences were observed comparing 500, 1000, 2000 and 4000 Hz before (T0) and after treatment (T1) (ANOVA: p <0.0001).” 

Based on the rest of the manuscript 3 kHz was used. Please correct.

We fixed the mistake.

(4)   Table 1. Please clarify in the Table header, why subjects are bolded.

We clarified this.

(5)   “Four patients (26.7%), were not satisfied about their hearing performance despite a recovery over 30, […]” 

Were the subjects not satisfied subjectively? Based on audiological outcome? Please elaborate.

We clarified this point.

(6)   Please provide outcome with CI.

We added the necessary info.

Discussion

(1)   “By reducing the thresholds of each frequency, stapedectomy statistically improved the PTAv also.” 

Please add “stapedectomy and hearing aids”.

Added

(2)   Please be careful in proposing conclusions from you study. 

We reduced the tone of our conclusions.

(3)   “One of the major complications of cochlear implant surgery in FAO is the stimulation of the facial nerve which is reported up to 38% [16,30,31].” 

Please give the range reported in the literature.

There was no range, it was a mistake. The sentence has been corrected as follow “One of the major complications of cochlear implant surgery in FAO is the stimulation of the facial nerve which was reported to be 38% by Rotteveel at al [23].”

(4)   “The short follow-up is a strong weakness in our study, as well as in other studies previously done [ref];” 

Please include all intended references [ref] before submitting or re-submitting a manuscript.

Done.

(5)   Adding the limitations of their study and possible future studies needed improves the manuscript greatly.

Thank you

Reviewer 2 Report

The manuscript has been extensively revised according to suggestions by the referees. There are still some questions. Both in the abstract as well as in the beginning of the Discussion it is mentioned that the results are preliminary. Why are these results preliminary? I also realize that new authors are added - why? And to which extent have the new authors contributed? The manuscript also needs an extensive English revision. After that process I recommend this manuscript for publication in Healthcare.

Author Response

Dear Reviewer,

Thank you for the thoughtful review of our manuscript and for sharing your valuable comments. We are grateful for the opportunity to submit a revised manuscript for your consideration. The critiques have allowed us to strengthen the manuscript and to address several areas requiring clarification. We have carefully read all the comments and revised the manuscript accordingly.

Below, we have responded to each comment point-by-point, presenting the reviewer critiques in regular type and our responses in bolded italics. Excerpts of revised text are also provided (indented text). In the main document, we have used track changes and red to show edits.

Thank you again for your time and insightful review of our paper

The manuscript has been extensively revised according to suggestions by the referees.

Thank you for your positive comment.

There are still some questions.

Both in the abstract as well as in the beginning of the Discussion it is mentioned that the results are preliminary. Why are these results preliminary?

We explained that “preliminary” was related to the small sample size

 I also realize that new authors are added - why? And to which extent have the new authors contributed?

Dr Ferlito is an expert neuro-otologist and supported us in the substantial revisions that we did and helped to re-analyze the data after new statistic was done.  We filled in the authors’ contribution section byn including this info.

The manuscript also needs an extensive English revision.

The English has been extensively revised.

After that process I recommend this manuscript for publication in Healthcare.

Thank you for your approval.